# COVID-19 symptom severity and duration among outpatients, July 2021-May 2023: The PROTECT observational study

Bhavya Vashi[1], Kristen Pettrone[1]*, Claire S. Wilson[2], Josh G. Chenoweth[1], Joost Brandsma[1], Melissa K. Gregory[1], Pavol Genzor[1], Deborah A. Striegel[1], Richard E. Rothman[2], Bhakti Hansoti[2], Gideon D. Avornu[2], Breana McBryde[2], Lauren Reynolds Zimmerman[2], Christopher W. Woods[3], Elizabeth A. Petzold[3], Jessica Cowden[4], Sorachai Nitayaphan[5], Worapong Nasomsong[6], Danielle V. Clark[1]

1 The Austere environments Consortium for Enhanced Sepsis Outcomes (ACESO), The Henry M. Jackson Foundation for the Advancement of Military Medicine, Inc., Bethesda, Maryland, United States of America, 2 Department of Emergency Medicine, Johns Hopkins University, Baltimore, Maryland, United States of America, 3 Division of Infectious Diseases, Department of Medicine, Duke University School of Medicine, Durham, North Carolina, United States of America, 4 Walter Reed Army Institute of Research-Armed Forces Research Institute of Medical Science (WRAIR-AFRIMS), Bangkok, Thailand, 5 Royal Thai Army, Armed Forces Research Institute of Medical Sciences, Bangkok, Thailand, 6 Division of Infectious Disease, Department of Internal Medicine, Phramongkutklao Hospital and College of Medicine, Bangkok, Thailand

* KPettrone@aceso-sepsis.org

## Abstract

### Introduction

With the emergence of new SARS-CoV-2 variants has come significant variations in disease manifestation, severity, and duration in non-hospitalized infected patients. To characterize symptom patterns and risk factors associated with symptom severity and duration, COVID-19 and influenza-like illness (ILI) outpatients and their contacts were enrolled at two sites in the United States of America and one site in Thailand.

### Methods

COVID-19 infection was confirmed at enrollment with a positive antigen or PCR test. Baseline demographics and medical histories were collected from participants at enrollment and daily self-reported symptom questionnaires were obtained to assess symptom severity and duration. Risk factors associated with symptom severity and duration were determined by multivariate logistic regression and Cox proportional hazards model.

### Results

Two hundred and forty one participants meeting the eligibility criteria were enrolled, including 174 confirmed COVID-19 cases (9% Delta and 90% Omicron), 33 ILI cases, and 34 healthy contacts. COVID-19 participants had a shorter median symptom duration of 9.0 (95% CI, 8.0–11.0) days than ILI participants. Infection with the Delta variant resulted in a longer symptom alleviation period compared to infection with the Omicron variant. The most commonly reported symptoms among COVID-19 participants were reported in the nasal

**Data Availability Statement:** De-identified data has been included as a supplemental file to this submission.

**Funding:** Funding for the Prophylaxis and Treatment of COVID-19 - Observational Study is executed by the Joint Program Executive Office for Chemical, Biological, Radiological and Nuclear Defense's (JPEO-CBRND) Joint Project Lead for CBRND Enabling Biotechnologies (JPL CBRND EB) on behalf of the Department of Defense's Chemical and Biological Defense Program awarded to KP and DC. This effort was in collaboration with the Defense Health Agency (DHA) COVID funding initiative for The Henry M. Jackson Foundation for the Advancement of Military Medicine, Inc. under award W911QY-20-9-0004 (2020 OTA) awarded to KP and DC. The funders had no role in study design, data collection and analysis, decision to publish, or preparation of the manuscript.

**Competing interests:** The authors have no competing interests.

and chest/respiratory domains of the FLU-PRO Plus. Participants infected with the Delta variant reported more symptoms overall, with significantly more symptoms affecting eyes and senses reported. 55% of SARS-CoV-2-positive participants reached a negative N1 Ct value by the day 14 study time point. No risk factors for moderate to severe symptoms were identified in this outpatient cohort. Male sex was associated with a shorter symptom duration.

## Conclusion

Symptom manifestation varied among Delta and Omicron variants. Few risk factors were identified for increased symptom severity or duration.

## Introduction

Severe acute respiratory syndrome coronavirus 2 (SARS-CoV-2) is a novel, pathogenic virus responsible for the coronavirus disease 2019 (COVID-19) pandemic affecting almost 775 million people (as of 4 Feb 2024) [1]. This highly transmittible respiratory pathogen is capable of mutating repeatedly into new variants of concern (VOC) and eluding therapeutic options. For example, mutations in the Delta and Omicron VOCs have led to changes in viral infectivity and transmissibility and resulted in decreased neutralization efficacy by monoclonal antibodies and vaccines [2]. Mutations may also lead to a failure of detection by molecular diagnostic tests and broad variations in clinical characteristics, including changing patterns of symptom manifestation, disease severity, and sensitivity to therapeutics [3]. As such, COVID-19 still is prevalent in society today and responsible for a significant burden of respiratory illness in the community. Pandemic and seasonal respiratory infections cause significant morbidity and mortality globally [4]. Acute upper and lower respiratory infections are associated with a large number of excess deaths and hospitalizations, but also outpatient visits as well as absences from work or school disrupting the social fabric of society [5].

Understanding the physiologic changes, host biomarker responses, and early disease viral burden of acute respiratory infections (ARI) such as SARS-CoV-2 can provide valuable prognostic and predictive information allowing for rapid and timely diagnosis of patients, improved clinical management, and minimize disruptions to everyday life [6, 7]. To date, characteristics of SARS-CoV-2 infected outpatients or their asymptomatic contacts have been described [8, 9], however, there is a gap with new emerging VOCs and changing patterns of disease. More up-to-date information is needed to better understand the development and progression of disease in outpatient settings following infection with SARS-CoV-2. Studying healthy close contacts of infected cases provides the opportunity to evaluate transmission patterns and capture the physiologic response to infection in the early stages of disease.

The pandemic has resulted in unprecedented efforts to rapidly develop and evaluate products to counter the impact of the virus. Approved anti-SARS-CoV-2 antiviral therapies decrease progression to severe disease in high risk individuals [10]. However, 50–60% of infections occur in standard risk outpatients with SARS-CoV-2 infection [11]. Currently, there are no approved therapies to shorten symptom duration in outpatients at standard risk for progression to severe disease [12]. The overarching goal of this observation cohort study was to generate data in support of the design of an adaptive clinical trials platform for therapeutic interventions for the treatment of mild to moderate COVID-19 illness. Uniquely, this observational study, Prophylaxis and Treatment of COVID-19—Observational Study

(PROTECT-Obs; ClinicalTrials.gov #: NCT04844541) [13], is running in parallel to a therapeutic platform trial (Prophylaxis and Treatment of COVID-19—Adaptive Platform Trial or PROTECT-APT; ClinicalTrials.gov #: NCT05954286) [14] evaluating promising investigational therapies for SARS-CoV-2 in the same outpatient study population. Data from this observational study is being used to inform design elements of PROTECT-APT, optimize study schedule, draw sample size assumptions, and evaluate biomarkers for potential incorporation into the platform trial. The primary objective of this study is to characterize the clinical, biological, virological, immunological, and pathological characteristics of patients with outpatient SARS-CoV-2 infection and other ARI among COVID-19 and influenza-like illness (ILI) cases and uninfected close contacts. This study also aims to demonstrate the use of novel technologies and modalities for the conduct of clinical trials during a pandemic including self-specimen collection [15], symptom self-reporting, remote physiologic monitoring, portable ultrasound [16], and virtual interaction with participants to enhance remote clinical trial capacity.

## Methods

### Study design

The PROTECT-Obs study received IRB approval by Advarra Institutional Review Board (MOD01484079). The study staff obtained written informed consent from each participant at the time of enrollment. Participants 18 years and older were recruited across three healthcare systems and shared or congregate living arrangements between July 15, 2021 and May 25, 2023. Outpatients with COVID-19 (diagnosed using antigen test and/or PCR) and their healthy contacts were enrolled at Duke University Hospital (Durham, North Carolina) from July 15, 2021 to February 22, 2022, Johns Hopkins University's Johns Hopkins Hospital and Bayview Medical Center (Baltimore, Maryland) from November 22, 2021 to May 25, 2023, and Phramongkutklao Hospital (Bangkok, Thailand) from October 17, 2022 to April 19, 2023. These sites were chosen due to higher SARS-CoV-2 prevalence, available research infrastructure, and to leverage existing collaborations. Beginning November 22, 2021, enrollment was expanded at Duke University Hospital and John Hopkins University to include ILI outpatients meeting the World Health Organization (WHO) ILI case definition (*"An acute respiratory illness with a measured temperature of ≥38°C and cough, with onset within the past 10 days"* [17]) and their contacts. Eligible participants were enrolled in the following cohorts: (a) symptomatic SARS-CoV-2 case with a positive antigen or PCR test and symptom onset within five days prior to enrollment (ten days prior to enrollment if enrolled between July 15, 2021 to May 21, 2021) (b) ILI case (c) SARS-CoV-2 contact or (d) ILI contact. Two hundred and fifty participants were screened for eligibility and, of these, 241 participants provided informed consent. Baseline demographics, comorbidities & risk factors, and vaccination status data were collected at the time of enrollment for these participants. The study schedule included both in-person and remote options. Participants were required to complete an in-person visit on day 0 but visit type (in person vs remote) varied on days 3, 7, 10, 14, 21, 28, and 90. Biospecimens including blood and nasal swabs were collected as part of the observational study. Venous blood draws, including collection of PAXgene blood RNA tubes, were performed at all in-person study visits. Participants were trained by study staff at day 0 to self-collect capillary blood using the Tasso Serum Separator (SST) blood collection device [15]. Tasso SSTs were collected at all study time points. Study staff collected nasal swabs at all in-person visits and training was provided to participants on self-nasal swab collection for remote visits. Additionally, point-of-care lung ultrasound was performed at in-person study visits [16]. Study data was stored using REDCap (Research Electronic Data Capture) [18].

## Influenza Patient-Reported Outcome (FLU-PRO) Plus

The validated FLU-PRO Plus Questionnaire was used for symptom ascertainment in our observational study. This instrument has been previously validated for influenza [19], and subsequently modified for COVID-19 with the addition of two questions assessing loss of taste and loss of smell [20]. The questionnaire measures the presence and severity of 32 symptoms across six symptom domains (nasal, throat, eyes, chest/respiratory, gastrointestinal, and body/ systematic). Participants rate their symptoms on a five-point Likert scale ranging from "not at all" to "very much", "never" to "always", or "0 times to 4 or more times". The FLU-PRO Plus questionnaire was collected daily until day 14 and then on days 21, 28, and 90 to assess symptom severity and duration. The questionnaire was distributed at midnight and participants were allowed 24 hours to complete.

## Respiratory testing

Nasopharyngeal swab specimens were collected by study staff from all participants at the time of enrollment (Day 0 visit) for respiratory testing. Specimens were run on the BioFire® Respiratory 2.1 (RP2.1) Panel using the BioFire® FilmArray® 2.0 system. The RP2.1 panel uses rapid multiplexed PCR (45 minutes) to simultaneously detect adenovirus, coronaviruses (229E, HKU1, NL63, OC43), SARS-CoV-2, human metapneumovirus, human rhinovirus/ enterovirus, influenza A (H1, H3, H1-2009), influenza B, parainfluenza viruses 1–4, respiratory syncytial virus (RSV), and bacteria including *Bordetella pertussis*, *Bordetella parapertussis*, *Chlamydophila pneumoniae*, and *Mycoplasma pneumoniae*.

For SARS-CoV-2 variant sequencing, RNA was extracted from 140 uL of nasopharyngeal swab universal transport medium by QIAamp Viral RNA Mini Kit. The DNA library was prepared using the Oxford Nanopore Technologies (ONT) Midnight SARS-CoV-2 protocol [21]. Sequencing was performed using the MinKNOW software and downstream analysis was carried out on EPI2ME Labs using the wf-artic bioinformatics workflow [22].

Nasal swabs collected from participants at Duke University Hospital and John Hopkins University at study time points were used for viral testing. The Centers for Disease Control and Prevention (CDC) real-time reverse transcription polymerase chain-reaction (rRT-PCR) panel was performed using the TaqPath 1-Step RT-qPCR Master Mix, CG (Thermo Fisher Scientific). The N1 probe cycle threshold (Ct) value was used as an indicator of viral burden.

## Statistical analysis

Outcomes of interest included symptom duration and symptom severity, based on FLU-PRO Plus responses. Symptom duration was evaluated as time from onset of symptoms to resolution. Symptomatic subjects had at least two or more symptoms rated moderate to severe at the time of enrollment in the FLU-PRO Plus questionnaire. Date of symptom onset was captured in a patient-reported symptom history form at enrollment and used to center the data. Resolution of symptoms was defined as the first of three consecutive days where all FLU-PRO Plus symptoms were reported as 0 ("not at all", "never", or "0 times") or 1 ("a little bit", "rarely", or "1 time"). Times to symptom resolution were dichotomized to 1–14 days or greater than 14 days. Median time to symptom resolution was calculated using Kaplan-Meier curves and survival analysis was performed to assess if symptom duration varied between COVID and ILI participants, Delta and Omicron variants, and sites. For symptom severity, participants were said to have "moderate to severe" symptoms if at least two symptoms on the FLU-PRO Plus were rated as a 3 ("quite a bit", "often", or "3 times") or 4 ("very much", "always", or "4 or more times") within the first five days post symptom onset. This was compared to participants who did not have two or more symptoms rated a 3 or 4 on the FLU-PRO Plus symptom

responses. Differences in symptom frequencies between groups were compared using Chi-square tests [23].

To identify demographic or clinical factors associated with symptom severity and duration, multivariate logistic regression and Cox proportional hazards models were performed, respectively [23, 24]. Both models included sex, age, race, vaccination, and comorbidities as covariates. The models were further adjusted by study site. Symptom duration analysis was limited to participants who completed the FLU-PRO Plus questionnaire on at least half the days. Participants who did not meet the symptom resolution definition while daily questionnaires were administered were censored at day 14.

All analyses were conducted in R, version 4.1.0 [23].

## Results

### Study population

We enrolled 241 participants under PROTECT-Obs between July 2021 and March 2023 (Fig 1). Among these participants, 174 had confirmed SARS-CoV-2 infections using the BioFire Respiratory Panel on enrollment nasopharyngeal swabs. A subset of COVID participants (n = 5) presented with a rhinovirus/enterovirus co-infection. Thirty-three symptomatic ILI participants were also evaluated using the BioFire panel; 11 tested positive for a viral infection including coronavirus OC43, metapneumovirus, rhinovirus/enterovirus, influenza A, parainfluenza virus 3, and RSV.

Sixty-one percent of the total enrolled participants were female, and the average age of the study population was 40 (Table 1). The majority of our participants were enrolled at Johns Hopkins University (56.4%), while enrollments in Thailand accounted for 30.3% of the total cohort. White and Black participants made up just over half of the population at 26.1% and 27.8%, respectively, with 38.2% identifying as Asian. Approximately half (48.1%) of the participants reported no comorbidities. Over 80% of participants documented at least two doses of a COVID-19 vaccine, with only about 10.3% of COVID-19 participants reporting no vaccination.

Variant testing found participants enrolled when PROTECT-Obs was initiated in July 2021 were predominantly infected with the Delta variant of SARS-CoV-2 (Fig 2). The first

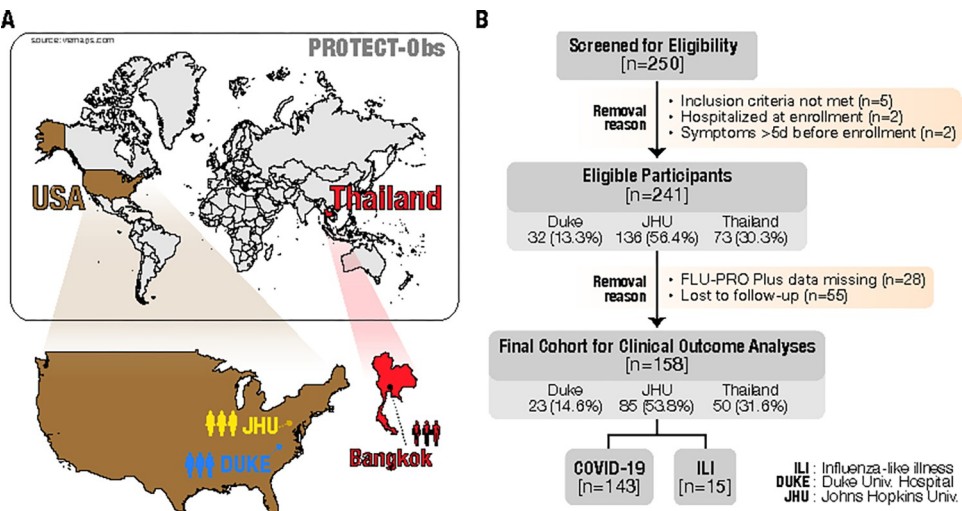

**Fig 1.** PROTECT-Obs study (A) sites and (B) screening/ enrollment of participants.

**Table 1. Baseline demographics and clinical characteristics of participants enrolled in the PROTECT-Obs study.**

| | COVID-19 | ILI | Healthy Contact | Overall |
|---|---|---|---|---|
| | (N = 174) | (N = 33) | (N = 34) | (N = 241) |
| **Sex at Birth** | | | | |
| Male | 65 (37.4%) | 10 (30.3%) | 16 (47.1%) | 91 (37.8%) |
| Female | 109 (62.6%) | 23 (69.7%) | 16 (47.1%) | 148 (61.4%) |
| Missing | 0 (0%) | 0 (0%) | 2 (5.9%) | 2 (0.8%) |
| **Age** | | | | |
| Mean (SD) | 39.7 (14.4) | 38.9 (15.7) | 43.4 (14.9) | 40.1 (14.6) |
| Median [Min, Max] | 37.0 [18.0, 84.0] | 35.0 [20.0, 73.0] | 43.0 [21.0, 75.0] | 37.0 [18.0, 84.0] |
| Missing | 0 (0%) | 0 (0%) | 1 (2.9%) | 1 (0.4%) |
| **Race** | | | | |
| White | 51 (29.3%) | 8 (24.2%) | 4 (11.8%) | 63 (26.1%) |
| Black or African American | 42 (24.1%) | 21 (63.6%) | 4 (11.8%) | 67 (27.8%) |
| Asian | 66 (37.9%) | 2 (6.1%) | 24 (70.6%) | 92 (38.2%) |
| Other | 15 (8.6%) | 2 (6.1%) | 1 (2.9%) | 18 (7.5%) |
| Missing | 0 (0%) | 0 (0%) | 1 (2.9%) | 1 (0.4%) |
| **Comorbidities or Risk Factors (#)** | | | | |
| None | 86 (49.4%) | 7 (21.2%) | 23 (67.6%) | 116 (48.1%) |
| 1 | 27 (15.5%) | 5 (15.2%) | 5 (14.7%) | 37 (15.4%) |
| 2 | 27 (15.5%) | 10 (30.3%) | 4 (11.8%) | 41 (17.0%) |
| 3 or more | 29 (16.7%) | 10 (30.3%) | 1 (2.9%) | 40 (16.6%) |
| Unknown | 5 (2.9%) | 1 (3.0%) | 1 (2.9%) | 7 (2.9%) |
| **COVID-19 Vaccination (# of doses)** | | | | |
| 2 or more | 146 (83.9%) | 19 (57.6%) | 30 (88.2%) | 195 (80.9%) |
| 1 | 6 (3.4%) | 3 (9.1%) | 1 (2.9%) | 10 (4.1%) |
| Unvaccinated | 18 (10.3%) | 11 (33.3%) | 1 (2.9%) | 30 (12.4%) |
| Unknown | 4 (2.3%) | 0 (0%) | 2 (5.9%) | 6 (2.5%) |
| **Study Site** | | | | |
| Duke | 25 (14.4%) | 3 (9.1%) | 4 (11.8%) | 32 (13.3%) |
| JHU | 99 (56.9%) | 30 (90.9%) | 7 (20.6%) | 136 (56.4%) |
| Thailand | 50 (28.7%) | 0 (0%) | 23 (67.6%) | 73 (30.3%) |

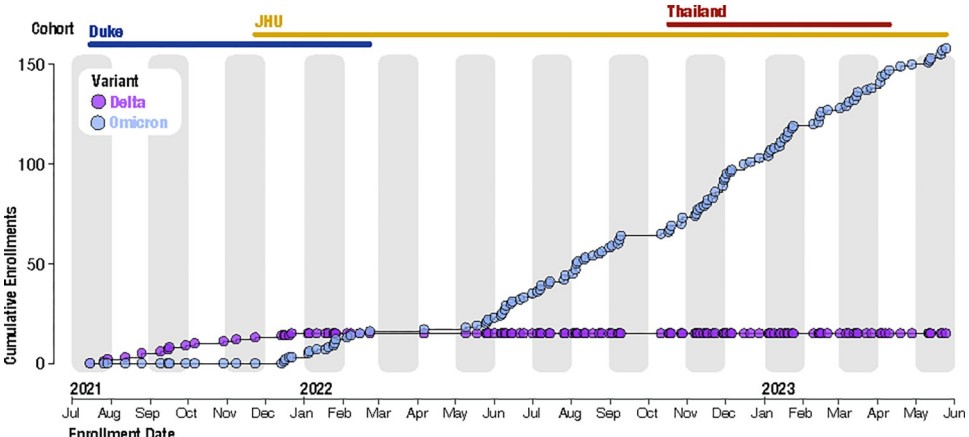

**Fig 2. Cumulative enrollments of Delta and Omicron variants among SARS-CoV-2-positive participants over study enrollment period.**

**Table 2. Variant sequencing of SARS-COV-2-positive participants at enrollment.**

| | Duke | JHU | Thailand | Overall |
|---|---|---|---|---|
| | (N = 25) | (N = 99) | (N = 50) | (N = 174) |
| **Variant** | | | | |
| Delta | 12 (48.0%) | 3 (3.0%) | 0 (0%) | 15 (8.6%) |
| Omicron | 12 (48.0%) | 94 (94.9%) | 50 (100%) | 156 (89.7%) |
| Other | 1 (4.0%) | 0 (0%) | 0 (0%) | 1 (0.6%) |
| Missing | 0 (0%) | 2 (2.0%) | 0 (0%) | 2 (1.1%) |

confirmed Omicron case in our study was identified in mid-December 2021. Nine percent of our SARS-CoV-2-positive participants were infected with the Delta variant and 90% were infected with the Omicron variant (Table 2). Additionally, one participant infected with the Iota variant was enrolled in July 2021.

## Symptomatology

**Symptom duration.** The median (95% confidence interval [CI]) symptom duration for the total study population, including COVID-19 and ILI participants, was 10.0 (9.0–12.0) days from symptom onset. COVID-19 participants had a median symptom duration of 9.0 (8.0–11.0) days which was significantly shorter than the median duration of 14.0 (12.0–17.0) days in ILI participants (Table 3). Of the subset of COVID-19 participants who completed the FLU-PRO Plus at day 90 (n = 34), 47% (n = 16) reported the presence of at least one symptom. Participants infected with the Delta variant had a longer symptom alleviation period of 14.0 (13.0–23.0) compared with those infected with Omicron who exhibited a median symptom duration of 9.0 (7.0–11.0) days. Geography was also associated with symptom duration. Thailand participants had an overall shorter symptom duration of 6.0 (5.0–9.0) days, compared to the US sites with a median symptom duration of 17.0 (13.0–22.0) days at Duke University and 12.0 (10.0–15.0) days at Johns Hopkins University.

**Symptom prevalence.** The most commonly reported symptoms, regardless of severity, among COVID-19 participants were those in the nasal (n = 151, 91.0%) and chest/respiratory (n = 153, 92.2%) domains of the FLU-PRO Plus questionnaire (Table 4). Chest/respiratory and body/systemic symptoms were most often reported as moderate to severe in COVID participants, 51.8% (n = 86) and 49.4% (n = 82) respectively, throughout the entirety of the FLU-PRO

**Table 3. Times to symptom resolution of study participants stratified by enrollment status, COVID-19 variant, and study site.**

| | N | Time to Resolution, Median No. of Days (95% CI) |
|---|---|---|
| **Enrollment Status** | | |
| COVID-19 | 129 | 9 (8, 11) |
| ILI | 12 | 14 (12, 17) |
| **Variant** | | |
| Delta | 11 | 14 (13, 23) |
| Omicron | 117 | 9 (7, 11) |
| **Site** | | |
| Duke | 22 | 17 (13, 22) |
| JHU | 69 | 12 (10, 15) |
| Thailand | 50 | 6 (5, 9) |

**Table 4. Proportions of COVID and ILI participants exhibiting symptoms of any severity or moderate to severe symptoms within the seven FLU-PRO Plus domains.**

| FLU-PRO Domain | Symptoms of Any Severity | | | Moderate to Severe Symptoms | | |
|---|---|---|---|---|---|---|
| | COVID (n = 166) | ILI (n = 28) | p-value | COVID (n = 166) | ILI (n = 28) | p-value |
| Nasal | 151 (91.0%) | 25 (89.3%) | 0.949 | 62 (37.3%) | 15 (53.6%) | 0.157 |
| Throat | 136 (81.9%) | 21 (75.0%) | 0.546 | 53 (31.9%) | 13 (46.4%) | 0.200 |
| Chest | 153 (92.2%) | 24 (85.7%) | 0.450 | 86 (51.8%) | 19 (67.9%) | 0.170 |
| Eyes | 82 (49.4%) | 18 (64.3%) | 0.210 | 21 (12.7%) | 8 (28.6%) | 0.058 |
| Body/Systematic | 142 (85.5%) | 23 (82.1%) | 0.857 | 82 (49.4%) | 16 (57.1%) | 0.580 |
| Gastrointestinal | 98 (59.0%) | 21 (75.0%) | 0.163 | 36 (21.7%) | 12 (42.9%) | 0.030 |
| Sensory | 59 (35.5%) | 10 (35.7%) | 1.000 | - | - | - |

P-values were calculated using Chi-square tests.

Plus collection period. ILI participants reported symptoms from similar FLU-PRO domains as those affected with COVID-19. Participants with ILI more frequently reported moderate to severe symptoms affecting the eyes (n = 8, 28.6%) or gastrointestinal systems (n = 12, 42.9%), compared to only 12.7% (n = 21) and 21.7% (n = 36) of those with COVID-19 (eyes: p-value, 0.058; gastrointestinal: p-value, 0.030).

The Delta variant was associated with symptoms more frequently affecting the eyes (p-value, 0.042) and senses (p-value, 0.001) when compared to Omicron (Table 5). Delta participants reported more symptoms overall, regardless of FLU-PRO Plus domain. No one variant exhibited a higher frequency of moderate to severe symptoms in any FLU-PRO Plus domain.

**Viral burden.** Of the 84 SARS-CoV-2-positive participants who had SARS-CoV-2 rRT-PCR data available, 55% reach the negative threshold of an N1 Ct value of 40 by the day 14 study time point. Cycle threshold values for COVID-19 participants with symptom resolution within 14 days (n = 14) fell at a similar pace when compared to participants with symptoms lasting longer than 14 days (n = 29) (Fig 3A). Participants reporting at least two or more moderate to severe symptoms (n = 30) initially had higher Ct values between 20 to 25, with a peak Ct value at the time of symptom onset, than those with mild to no symptoms (n = 23) whose Ct values were between 25 to 30 around 5 to 7 days post symptom onset (Fig 3B). However, this divergence did not persist throughout the course of the disease, with both groups reaching similar Ct values around day 10 post symptom onset.

**Table 5. Proportions of Delta and Omicron COVID-19 participants exhibiting symptoms of any severity or moderate to severe symptoms within the seven FLU-PRO Plus domains.**

| FLU-PRO Domain | Symptoms of Any Severity | | | Moderate to Severe Symptoms | | |
|---|---|---|---|---|---|---|
| | Delta (n = 14) | Omicron (n = 149) | p-value | Delta (n = 14) | Omicron (n = 149) | p-value |
| Nasal | 14 (100%) | 134 (89.9%) | 0.446 | 4 (28.6%) | 57 (38.3%) | 0.669 |
| Throat | 13 (92.9%) | 122 (81.9%) | 0.502 | 4 (28.6%) | 48 (32.2%) | 1.000 |
| Chest | 14 (100%) | 136 (91.3%) | 0.525 | 8 (57.1%) | 76 (51.7%) | 0.911 |
| Eyes | 11 (78.6%) | 69 (46.3%) | 0.042 | 0 (0%) | 20 (13.4%) | 0.299 |
| Body/Systematic | 14 (100%) | 125 (83.9%) | 0.218 | 9 (64.3%) | 71 (47.7%) | 0.362 |
| Gastrointestinal | 8 (57.1%) | 87 (58.4%) | 1.000 | 2 (14.3%) | 34 (22.8%) | 0.690 |
| Sensory | 11 (78.6%) | 47 (31.5%) | 0.001 | - | - | - |

P-values were calculated using Chi-square tests.

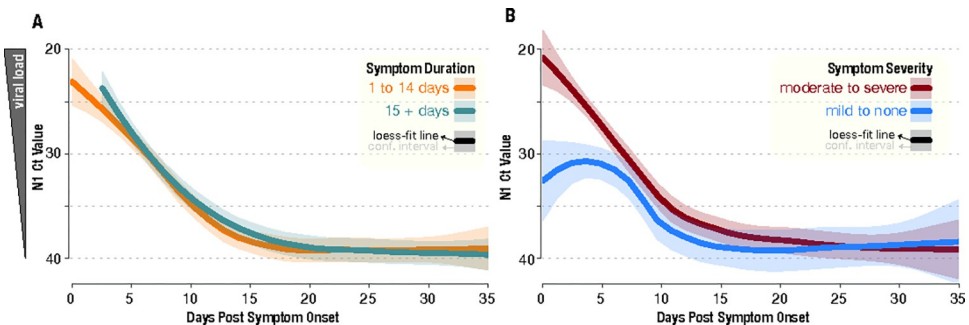

**Fig 3.** N1 cycle threshold (Ct) values after symptom onset stratified by dichotomous (A) symptom duration and (B) symptom severity outcomes.

## Clinical outcomes

For symptom evaluation, we limited analyses to participants who completed a minimum of half the FLU-PRO Plus questionnaires. This resulted in a subset of 158 patients, 143 of whom tested positive for SARS-CoV-2 on the BioFire® Respiratory 2.1 (Fig 1).

To evaluate potential associations between various participant characteristics and disease severity, multivariate logistic regression was performed. Overall, out of the SARS-CoV-2-positive participants who completed at least 50% of FLU-PRO Plus questionnaires, 60.8% reported at least two symptoms as moderate to severe in the first five days of symptom onset. The adjusted odds of presenting with moderate to severe symptoms were not statistically significant for male sex (adjusted odds ratio [aOR], 0.48; 95% CI, 0.18 to 1.23), race (Black or African American: aOR, 2.05; 95% CI, 0. 36 to 16.67; Asian: aOR, 0.84; 95% CI, 0.13 to 6.62; Other: aOR, 0.35; 95% CI, 0.05 to 2.36 compared to White participants), vaccination (aOR, 1.07; 95% CI, 0.12 to 7.53), or comorbidities (one comorbidity: aOR, 0.40; 95% CI, 0.08 to 1.67; two comorbidities: aOR, 2.32; 95% CI, 0.55 to 11.68; three or more comorbidities: aOR, 2.77; 95% CI, 0.50 to 18.92 compared to those with no comorbidities) (Fig 4A). Older age was associated with less severe symptoms in this outpatient cohort (aOR, 0.24; 95% CI, 0.08 to 0.70).

A Cox proportional hazards model was fit to identify predictors of symptom duration using the same covariates and study site adjustment. Age (adjusted hazards ratio [aHR], 0.73; 95% CI, 0.44 to 1.23), race (Black or African American: aHR, 0.75; 95% CI, 0.31 to 1.81; Asian: aHR, 0.41; 95% CI, 0.14 to 1.17; Other: aHR, 1.38; 95% CI, 0.40 to 4.79 compared to White participants), vaccination (aHR, 0.72; 95% CI, 0.29 to 1.82), or comorbidities (one comorbidity: aHR, 1.63; 95% CI, 0.86 to 3.08; two comorbidities: aHR, 0.84; 95% CI, 0.36 to 1.95; three or more comorbidities: aHR, 0.82; 95% CI, 0.34 to 2.02 compared to those with no comorbidities) were not found to be associated with a prolonged symptom duration (Fig 4B). However, male sex was associated with a shorter symptom duration (aHR, 1.75; 95% CI, 1.06 to 2.88).

## Discussion

Gaps in understanding of COVID-19 disease severity and duration are further complicated by emerging VOCs and changing patterns of disease over the course of the pandemic. Our study captures longitudinal symptomatologic data from multiple variants of SARS-CoV-2 from 2021–2023. Our findings of the timing of the emergence of the Omicron variant in mid-December 2021 in our study population are consistent with other reports of temporal and geographic predominance of the Delta and Omicron variants [25, 26].

While many previous studies focus on the risk of severe outcomes from COVID-19 infection including hospitalization and death, there are few published papers evaluating the risk of

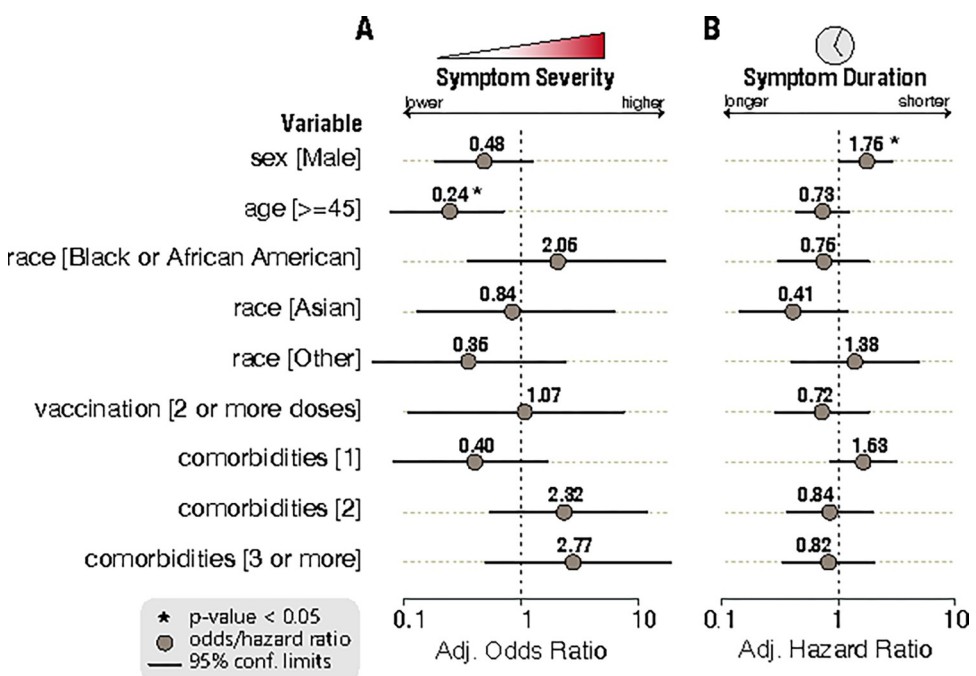

**Fig 4.** (A) Odds of moderate to severe symptom severity in a multivariate logistic regression model and (B) hazards ratios of longer symptom durations in a multivariate Cox proportional hazards model. Study site was included in the models to adjust for potential heterogeneity.

increased COVID-19 symptom severity in an outpatient cohort. Age and presence of comorbidities have been previously shown to be predictive of adverse events among outpatients [27]. In our study, there was no increased risk of severe symptoms by race or among participants with comorbidities. Older age was associated with less severe symptoms in our cohort and was also not associated with longer symptom durations. Older age has been linked to prolonged illness in other inpatient and outpatient studies which has been attributed to factors such as immunosenescence and inflammaging [28–30]. While we did not observe similar associations of disease severity with age, our smaller sample size and exclusion of hospitalized participants may have impacted our ability to detect such relationships.

While vaccination against SARS-CoV-2 has been demonstrated to protect against SARS-CoV-2 infection and decrease COVID-19-associated hospitalization and death, little is known about its impact on symptom severity reduction in outpatients [31, 32]. We found no association between vaccination and protection against moderate to severe symptom severity, although low numbers of unvaccinated participants may have contributed to this finding. While several participants in our study had co-infections that may have impacted the COVID-19 severity and symptom duration, the small number of co-infections prevented an evaluation of this effect.

For symptom duration, age, vaccination, and the presence of comorbidities did not increase the risk for prolonged illness in this outpatient cohort. The different methodology for defining symptom duration in an outpatient cohort with remote data collection may explain some of these differences. Race was not associated with symptom duration in this model; however, this may be due to the additional adjustment by site as a majority of our Asian participants were enrolled in Thailand. Male participants had shorter symptom durations, which is consistent with another outpatient study [29]. This may be attributed to sex-based differences in viral load decay speeds that have been seen in other studies where male gender was associated with

faster decay speeds with some VOC infections and baseline viral loads compared to females [33].

Overall, symptom duration of COVID-19, especially for more recent variants, is not well delineated. Early in the pandemic in 2020, symptom duration ranged from 6–67 days [34] with medians of 15 [8], 16 [28], and 18 [34] days. In our cohort, we found a similar duration for the Delta variant (14 days), with a shortening duration for the Omicron variant (9 days). However, it is estimated that up to 80% of infected patients develop at least one long-term symptom [35]. Although the study was not designed to evaluate post-acute COVID syndrome (PACS), 47% of our participants reported the presence of at least one symptom at day 90 out of those who completed the symptom questionnaire.

Outside of the characteristics described in the study, additional factors may contribute to the differences in symptom manifestation, severity and duration observed in our study population. Mutations present in the Delta and Omicron variants have resulted in alterations in tissue tropism between the two VOCs. Omicron has been shown to replicate faster in nasal epithelial cells and Delta replicated faster in the lungs [36]. This may be due to Omicron's TMPRSS-independent viral entry mechanism that broadens the tissue susceptibility to the variant [37]. This may account for the differences in symptom manifestation seen in ours and other studies where upper respiratory symptoms were more common in individuals infected with the Omicron variant [38]. Host factors also play a role in disease severity and symptom manifestation. Prior infection with SARS-CoV-2, variations in the individual host response and differences in host genetics, such as TMPRSS polymorphisms, may dampen or augment disease severity [39]. Similarly, the SARS-CoV-2 virus exhibits a direct effect on the host immune response. The Delta variant stimulates gene expression of pro-inflammatory cytokines, such as interferon-gamma, which may account for the greater magnitude of systemic systems seen in ours and other studies [38, 40]. There are limitations to be addressed in our study. First, our outcomes are based on self-reported data. We chose to use the FLU-PRO Plus, which is a validated instrument that allows us to determine symptom severity and presence, as well as a global assessment of the infection. Symptom severity as measured by the FLU-PRO Plus has also previously been shown to be correlated with CRP and IP-10, clinical biomarkers associated with disease severity, in SARS-CoV-2 and ILI patients [41]. However, we did note a high drop-out rate with COVID-19 and ILI participants not completing the full FLU-PRO Plus series. This may introduce bias with patients with more severe symptoms choosing to fill out the questionnaire on any given day or those with prolonged symptoms continuing to complete the entire series. Additionally, we did not initially limit symptom onset to five days. This resulted in participants enrolled with longer symptom onsets, at which point they may test negative for infection or may have resolved or milder symptoms. We controlled for this by shifting all symptom durations to reflect symptom onset dates provided by participants. Moreover, defining "symptom resolution" was challenging due to the fluctuating nature of COVID-19 symptoms [9], which may resolve only to recur or improve and subsequently increase. To attempt to address this, we chose to define symptom resolution as meeting the definition for three consecutive days. Finally, site differences, particularly variant, race, and comorbidities, may have impacted our results. Participants infected with the Delta variant were primarily enrolled at Duke University Hospital, with only a few participants enrolled at Johns Hopkins University. Thailand participants were generally healthier, reporting fewer comorbidities and higher vaccination rates than the US sites, and accounted for a majority of the Asian enrollments. Other factors such as racial variations in immune response to infection and vaccination may have impacted symptom duration with several studies reporting stronger cellular and humoral immune response and longer duration of immunity post vaccination and native infection in Asian race compared to White race [42, 43]. Site was therefore added as a covariate to the models.

The PROTECT-Obs study allowed us to use a data-driven approach to design the therapeutic platform trial PROTECT-APT. We were able to design an adaptive trial that maximized potential for quantifying effectiveness and efficacy, but restricted study procedures to ensure they were high-yield and feasible. Investigators from both studies were able to evaluate symptom pattern and duration in study participants which enabled the utilization of a more simplified symptom diary for the platform trial. Data from the observation study provided reliable estimates for assumptions used in the calculation of the sample size for the platform trial's master protocol symptom resolution primary endpoint. The observation study also trialed varying study activity schedules, remote assessments and biospecimen sampling techniques which allowed for the optimization of these activities in the clinical trial.

A majority of COVID-19 patients do not require hospitalization, yet there remains a lack of information on the clinical characteristics of outpatients with mild to moderate disease. The shift in focus during the pandemic to provide remote care resulted in an increased capacity for conducting remote clinical trials. In our study, we were able to capture valuable data from a fully outpatient cohort to include multiple VOCs over the duration of the pandemic. More broadly and post-pandemic, findings from our study can assist with the understanding of the evolution of respiratory virus symptom patterns and severity over the course of an outbreak and with continued viral mutation. Our findings also contribute insight to the impact of factors such as demographics, geography and introduction of vaccination on disease severity in outpatients infection with novel viral respiratory pathogens.

## Supporting information

**S1 Table. Multivariate logistic regression model.**
(DOCX)

**S2 Table. Multivariate cox proportional hazards model.**
(DOCX)

**S1 Dataset.**
(XLSX)

## Acknowledgments

We thank the ACESO members (Qianru Wu, Debbie Lund, Margaret Farrell, Suzanne Restrepo, and Ramy Navas) for supporting the PROTECT-Obs study. We also thank the study team members at Duke University Hospital, Johns Hopkins University's Johns Hopkins Hospital and Bayview Medical Center (Rabbiya Iqbal, Liam Pauli, Nate Grigsby-Rocca, Querino Maia, Chase Yonamine, Ossama Saeed, Isai Ramirez-Gonzalez, Allison Jacobi-Dorbeck, Crystal LaBozzetta, Daniella Albanese, Samantha Bott, Laiyana Kabir, Michael Kramer, Mishel Malik, Joshua Rosario, Danielle Zambito, Jonathan Bearden, Anna Ross, Joshua East, Kevin Michael Lolies) and Phramongkutklao Hospital for their contribution to patient recruitment and obtaining clinical data and samples.

## Author Contributions

**Conceptualization:** Bhavya Vashi, Kristen Pettrone.

**Data curation:** Joost Brandsma.

**Formal analysis:** Josh G. Chenoweth, Joost Brandsma, Pavol Genzor, Deborah A. Striegel.

**Funding acquisition:** Kristen Pettrone, Danielle V. Clark.

**Investigation:** Bhavya Vashi, Kristen Pettrone, Claire S. Wilson, Josh G. Chenoweth, Joost Brandsma, Melissa K. Gregory, Deborah A. Striegel, Richard E. Rothman, Bhakti Hansoti, Sorachai Nitayaphan, Worapong Nasomsong, Danielle V. Clark.

**Methodology:** Kristen Pettrone, Joost Brandsma, Pavol Genzor, Deborah A. Striegel, Richard E. Rothman, Bhakti Hansoti, Danielle V. Clark.

**Project administration:** Gideon D. Avornu, Breana McBryde, Lauren Reynolds Zimmerman, Elizabeth A. Petzold.

**Supervision:** Bhavya Vashi, Kristen Pettrone, Josh G. Chenoweth, Bhakti Hansoti, Christopher W. Woods, Jessica Cowden, Sorachai Nitayaphan, Worapong Nasomsong, Danielle V. Clark.

**Writing – original draft:** Kristen Pettrone.

**Writing – review & editing:** Bhavya Vashi, Kristen Pettrone, Richard E. Rothman, Bhakti Hansoti, Christopher W. Woods, Jessica Cowden, Danielle V. Clark.

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
