## [Decision Letter · Decision Letter 0]

29 Sep 2024

PONE-D-24-20525COVID-19 symptom severity and duration among outpatients, July 2021-May 2023: The PROTECT observational studyPLOS ONE

Dear Dr. Pettrone,

Thank you for submitting your manuscript to PLOS ONE. After careful consideration, we feel that it has merit but does not fully meet PLOS ONE’s publication criteria as it currently stands. Therefore, we invite you to submit a revised version of the manuscript that addresses the points raised during the review process.

Editor's comments: 1. Line 62-65: "Understanding the physiologic changes, host biomarker responses, and early disease viral burden of acute respiratory infections (ARI) such as SARS-CoV-2 can provide valuable prognostic and predictive information allowing for rapid and timely diagnosis of patients, improved clinical management, and minimize disruptions to everyday life.": there are no references cited to support this claim. More references are suggested to be cited, with the following one as example (citing is optional): Liu BM, Hill HR. Role of Host Immune and Inflammatory Responses in COVID-19 Cases with Underlying Primary Immunodeficiency: A Review. J Interferon Cytokine Res. 2020 Dec;40(12):549-554. doi: 10.1089/jir.2020.0210. PMID: 33337932; PMCID: PMC7757688. 2. The RT-PCR for SARS-CoV-2 using ct value cannot provide viral load. Please avoid using "viral load" in this context as ct value is insufficient to provide viral load and PCR only detects RNA rather than viruses. Also, line 155, "N1 probe cycle threshold (Ct) value was used as an indicator of viral load." should be reworded ad this is inaccurate statement. Unless running standards, ct value can not predict viral load.

We look forward to receiving your revised manuscript.

Kind regards,

Benjamin M. Liu, MBBS, PhD, D(ABMM), MB(ASCP)

Academic Editor

PLOS ONE

Journal requirements: 1. When submitting your revision, we need you to address these additional requirements.Please ensure that your manuscript meets PLOS ONE's style requirements, including those for file naming. The PLOS ONE style templates can be found at https://journals.plos.org/plosone/s/file?id=wjVg/PLOSOne_formatting_sample_main_body.pdf and https://journals.plos.org/plosone/s/file?id=ba62/PLOSOne_formatting_sample_title_authors_affiliations.pdf 2. We note that the grant information you provided in the ‘Funding Information’ and ‘Financial Disclosure’ sections do not match.  When you resubmit, please ensure that you provide the correct grant numbers for the awards you received for your study in the ‘Funding Information’ section. 3. Thank you for stating the following financial disclosure:  [Funding for the Prophylaxis and Treatment of COVID-19 - Observational Study is executed by the Joint Program Executive Office for Chemical, Biological, Radiological and Nuclear Defense’s (JPEO-CBRND) Joint Project Lead for CBRND Enabling Biotechnologies (JPL CBRND EB) on behalf of the Department of Defense’s Chemical and Biological Defense Program. This effort was in collaboration with the Defense Health Agency (DHA) COVID funding initiative for The Henry M. Jackson Foundation for the Advancement of Military Medicine, Inc. under award W911QY-20-9-0004 (2020 OTA). ].  Please state what role the funders took in the study.  If the funders had no role, please state: ""The funders had no role in study design, data collection and analysis, decision to publish, or preparation of the manuscript."" If this statement is not correct you must amend it as needed. Please include this amended Role of Funder statement in your cover letter; we will change the online submission form on your behalf. 4. Please include a caption for figure 5. 5. In the online submission form, you indicated that [De-identified data may be made available upon reasonable request to the corresponding author.]. All PLOS journals now require all data underlying the findings described in their manuscript to be freely available to other researchers, either 1. In a public repository, 2. Within the manuscript itself, or 3. Uploaded as supplementary information.This policy applies to all data except where public deposition would breach compliance with the protocol approved by your research ethics board. If your data cannot be made publicly available for ethical or legal reasons (e.g., public availability would compromise patient privacy), please explain your reasons on resubmission and your exemption request will be escalated for approval. 

Reviewers' comments:

Reviewer's Responses to Questions

**Comments to the Author**

1. Is the manuscript technically sound, and do the data support the conclusions?

Reviewer #1: Yes

Reviewer #2: Partly

2. Has the statistical analysis been performed appropriately and rigorously? 

Reviewer #1: Yes

Reviewer #2: Yes

3. Have the authors made all data underlying the findings in their manuscript fully available?

Reviewer #1: Yes

Reviewer #2: Yes

4. Is the manuscript presented in an intelligible fashion and written in standard English?

Reviewer #1: Yes

Reviewer #2: Yes

5. Review Comments to the Author

Reviewer #1: My assessment / comments are as follows:

1. Is the manuscript technically sound, and do the data support the conclusions?

Yes

2. Has the statistical analysis been performed appropriately and rigorously?

Yes, extensive and appropriate statistical analyses have been performed and reported.

3. Have the authors made all data underlying the findings in their manuscript fully available?

Yes, de-identified data can be obtained from the corresponding author.

4. Is the manuscript presented in an intelligible fashion and written in standard English?

Yes

5. Review comments to the Author:

a. Page 2, Abstract:

• Line 30: I suggest that the authors use “United States of America” instead of just United States.

• Line 37: 9% Delta and 90% Omicron. What constituted the other 1%?

• Lines 36, 43: Starting a sentence with a number (241 participants; 55% of SARS-CoV-2-positive) makes reading difficult. The sentence needs to be reconstructed or figures should be written out in full. This also occurs elsewhere in the manuscript.

b. Page 5, Line 99: What was the reason / rationale for selecting this specific timeframe (June 15, 2021 to May 25, 2023)? Was this purely convenience sampling or was there any clinical relevance attached to the different periods at different sites?

c. Pages 5 & 6, Lines 111, 112: Why was enrolment window different for cases collected during 2021 (“ten days if enrolled between July 15, 2021 to May 21, 2021”)? Are these dates correct, since it is not in chronological order?

d. Page 6, Lines 120, 121: What is the volume of blood collected with the Tasso SST blood collection device?

e. Page 9, Lines 184, 185, 195: Error! Reference source not found. Tis is also the case in several other places in the manuscript.

f. Page 10, Lines 199, 200: What type of vaccines were received by those individuals reporting that they had been vaccinated? Would you expect a difference in the number and severity of symptoms between recipients of mRNA, vector and protein subunit vaccines?

g. Page 17, Lines 326, 327: What measures were put in place to avoid or limit bias in the self-reporting symptoms? Is it possible that (recall and survivor, amongst other) bias could have been the reason for the absence of an association between symptom severity and any of the other parameters measured?

h. When comparing cohorts from vastly different geographical and demographical perspectives, such as the USA and Thailand, it should be justified why the authors think the two countries are comparable. Besides the predominance of different VOCs circulating in the USA and Thailand, there are additional factors that could have played a role, such as population densities and other cultural practices. Was this considered when selecting the sites?

i. Finally, as part of the Discussion, I would have liked to see a section on how the current study can contribute to respiratory infections and illnesses post-COVID-19 pandemic.

I found this manuscript insightful and of high quality, but feel that addressing the above questions will enhance the value of the research. It is a pity about the references not being found, as it made it difficult to properly evaluate the facts.

Reviewer #2: The manuscript presents interesting results however, I feel like some context and correlations that should be included are missing.

1. In line 26 (Introduction): While the paper mentions the emergence of new SARS-CoV-2 variants, there is insufficient exploration of the virological mechanisms behind immune evasion, viral mutations, or changes in symptomatology. Expanding this section to discuss the specific mutations in the Delta and Omicron variants that contribute to altered viral fitness, transmission, and immune evasion would strengthen the context for the study.

2. In line 72 (Therapeutics Discussion): The paper briefly mentions the role of therapeutics in reducing COVID-19 severity but does not offer detailed, up-to-date information on the efficacy of antiviral treatments against the variants studied. Including recent findings on how treatments (e.g., monoclonal antibodies, antivirals) interact with variant-specific mutations would add depth to the study’s clinical implications.

3. In line 150 (SARS-CoV-2 Sequencing): The variant sequencing methods are sound, but the study lacks detailed integration between the virological data and clinical outcomes. Providing more information on how specific mutations in Delta and Omicron variants (e.g., spike protein changes) align with observed differences in symptom severity and duration would significantly enhance the methodological clarity.

4. In line 183 (Study Population Results): The study mentions co-infections with rhinovirus/enterovirus but does not explore how these co-infections might have affected SARS-CoV-2 symptomology or severity. Adding virological discussions on viral interference, co-infection dynamics, or immune modulation in the presence of multiple pathogens would provide a more complete understanding of the results.

5. In line 206 (Variant Testing): The paper describes the distribution of Delta and Omicron variants but lacks in-depth analysis of how variant-specific mutations contributed to the observed differences in symptom profiles. Discussing key mutations (e.g., in the spike protein or other viral components) that affect viral replication rates, immune evasion, or tissue tropism would improve the interpretation of these findings.

6. In line 218 (Symptomatology Results): Although the symptom data is presented for COVID-19 and ILI participants, there is no virological explanation for the observed differences. Including a discussion on how viral load kinetics, host immune response, and variant-specific replication dynamics might influence these differences would enrich the analysis.

7. In line 253 (Symptom Prevalence by Variant): The result that Delta was associated with more eye and sensory symptoms than Omicron requires further virological context. The paper could benefit from exploring how viral mutations in Delta potentially alter tissue tropism (e.g., greater affinity for conjunctival cells or sensory neurons) compared to Omicron, thereby leading to these symptom differences.

8. In line 264 (Viral Load Results): While viral load data is provided, the lack of correlation between viral load and symptom duration is not fully explored. A deeper discussion on variant-specific replication rates, immune responses (e.g., cytokine profiles), or viral persistence might explain why viral load did not show a significant association with symptom outcomes. Highlighting whether certain mutations in Delta or Omicron influence viral clearance rates could improve the understanding of the data.

9. In line 268 (Symptom Severity and Viral Load): The divergence in Ct values based on symptom severity is mentioned, but the underlying virological reasons are not discussed. The paper could be strengthened by addressing how early immune responses (e.g., interferon levels) or viral replication dynamics in specific tissues contribute to symptom severity in different SARS-CoV-2 variants.

10. In line 283 (Multivariate Logistic Regression on Symptom Severity): The paper found no significant association between demographic factors and symptom severity, but does not delve into potential virological explanations. Exploring how host-pathogen interactions, such as the variability in immune response based on age, race, or comorbidities, might influence disease outcomes for different variants would add depth to the interpretation.

11. In line 295 (Cox Proportional Hazards for Symptom Duration): The shorter symptom duration in males lacks a virological or immunological explanation. Discussing sex-based differences in immune response (e.g., T-cell activation, antibody production) or differences in viral clearance rates would improve the scientific rationale behind this observation. Additionally, integrating data on how viral kinetics might differ between sexes for the studied variants would provide more robust conclusions.

12. In line 342 (Long-Term Symptoms): The study mentions the presence of long-term symptoms in participants, but does not explore the virological mechanisms that might contribute to post-acute COVID syndrome. A discussion on viral persistence, chronic inflammation, or immune dysregulation could better explain the high percentage of patients reporting symptoms after 90 days, particularly in relation to different SARS-CoV-2 variants.

13. In line 355 (Symptom Fluctuations): The study addresses the fluctuating nature of symptoms but does not provide a virological explanation for these observations. Discussing how viral replication in different tissues or immune system reactivation might lead to fluctuating symptom patterns would offer a more comprehensive understanding of the disease course.

14. In line 365 (Site Differences): Although site differences in symptom duration are noted, the paper could benefit from more detailed analysis on how environmental factors, regional viral strains, or differences in immune responses among populations could contribute to these differences. A virological comparison of viral fitness or immune escape between regions might also clarify the observed outcomes.

6. PLOS authors have the option to publish the peer review history of their article (what does this mean?). If published, this will include your full peer review and any attached files.

Reviewer #1: No

Reviewer #2: No

---

## [Author Response · Author response to Decision Letter 0]

1 Nov 2024

The authors have revised the manuscript addressing all of the reviewers comments and uploaded the following files on Oct 22nd: 

1) Response to Reviewers

2) Revised manuscript with track changes

3) Manuscript (22 Oct)

31 Oct 2024. Per the Editor's request. A de-identified data set (.XLS format) has been uploaded as supplemental information.

---

## [Decision Letter · Decision Letter 1]

12 Nov 2024

COVID-19 symptom severity and duration among outpatients, July 2021-May 2023: The PROTECT observational study

PONE-D-24-20525R1

Dear Dr. Pettrone,

We’re pleased to inform you that your manuscript has been judged scientifically suitable for publication and will be formally accepted for publication once it meets all outstanding technical requirements.

Kind regards,

Benjamin M. Liu, MBBS, PhD, D(ABMM), MB(ASCP)

Academic Editor

PLOS ONE

Additional Editor Comments (optional):

Reviewers' comments:

Reviewer's Responses to Questions

**Comments to the Author**

1. If the authors have adequately addressed your comments raised in a previous round of review and you feel that this manuscript is now acceptable for publication, you may indicate that here to bypass the “Comments to the Author” section, enter your conflict of interest statement in the “Confidential to Editor” section, and submit your "Accept" recommendation.

Reviewer #1: All comments have been addressed

2. Is the manuscript technically sound, and do the data support the conclusions?

Reviewer #1: Yes

3. Has the statistical analysis been performed appropriately and rigorously? 

Reviewer #1: Yes

4. Have the authors made all data underlying the findings in their manuscript fully available?

Reviewer #1: Yes

5. Is the manuscript presented in an intelligible fashion and written in standard English?

Reviewer #1: Yes

6. Review Comments to the Author

Reviewer #1: Well done on the revised manuscript. You have addressed all comments and queries in a logical and concise manner and improved the manuscript significantly.

7. PLOS authors have the option to publish the peer review history of their article (what does this mean?). If published, this will include your full peer review and any attached files.

Reviewer #1: **Yes: **Prof Corena de Beer

---

## [Editor Report · Acceptance letter]

2 Dec 2024

PONE-D-24-20525R1 

PLOS ONE

Dear Dr. Pettrone, 

I'm pleased to inform you that your manuscript has been deemed suitable for publication in PLOS ONE. Congratulations! Your manuscript is now being handed over to our production team.

Kind regards, 

on behalf of

Dr. Benjamin M. Liu 

Academic Editor

PLOS ONE